# Thermally stable Ni foam-supported inverse CeAlO$_x$/Ni ensemble as an active structured catalyst for CO$_2$ hydrogenation to methane

Xin Tang[1,2,4], Chuqiao Song[1,2,4], Haibo Li[1,2], Wenyu Liu[1,2], Xinyu Hu[1], Qiaoli Chen [1], Hanfeng Lu[1], Siyu Yao [3] ✉, Xiao-nian Li[1,2] & Lili Lin [1,2] ✉

Nickel is the most widely used inexpensive active metal center of the heterogeneous catalysts for CO$_2$ hydrogenation to methane. However, Ni-based catalysts suffer from severe deactivation in CO$_2$ methanation reaction due to the irreversible sintering and coke deposition caused by the inevitable localized hotspots generated during the vigorously exothermic reaction. Herein, we demonstrate the inverse CeAlO$_x$/Ni composite constructed on the Ni-foam structure support realizes remarkable CO$_2$ methanation catalytic activity and stability in a wide operation temperature range from 240 to 600 °C. Significantly, CeAlO$_x$/Ni/Ni-foam catalyst maintains its initial activity after seven drastic heating-cooling cycles from RT to 240 to 600 °C. Meanwhile, the structure catalyst also shows water resistance and long-term stability under reaction condition. The promising thermal stability and water-resistance of CeAlO$_x$/Ni/Ni-foam originate from the excellent heat and mass transport efficiency which eliminates local hotspots and the formation of Ni-foam stabilized CeAlO$_x$/Ni inverse composites which effectively anchored the active species and prevents carbon deposition from CH$_4$ decomposition.

Ni-based catalysts are widely applied in the industrial CO methanation reaction and have shown great potential for the conversion of CO$_2$ to CH$_4$ (also known as the Sabatier reaction), due to the relatively high activity, selectivity and affordability[1–4]. Considering the potential application of CO$_2$ methanation in the integrated power-to-gas process containing CO$_2$ capture, renewable energy-powered hydrogen production (e.g., electrolysis of water) and CO$_2$ utilization modules, the development of active and durable Ni-based CO$_2$ methanation catalyst is highly desirable and urgently demanded[5–7].

Despite the promising perspective, it is challenging to apply conventional Ni/oxide catalysts in CO$_2$ methanation reactions. One of the problems is the formation of localized hotspots in the catalyst bed caused by the severe reaction heat and the relatively poor heat

conductivity of oxide hosting materials[8–10]. According to the literature estimation, an adiabatic temperature rise around 59.2 °C will be presented for each 1 mol% conversion of CO$_2$ in the hydrogenation reaction, and the adiabatic temperature can reach the maxima of 600 °C, the temperature corresponding to the thermal balance between exothermic CO$_2$ methanation and endothermic reverse water gas shift reactions[11,12]. Once the thermal disturbance exceeds the binding energy between Ni nanoparticle and support, the well-dispersed Ni species tend to migrate on the catalyst and further agglomerate into large particles driven by the surface energy[13,14]. What's more, the high concentration of steam in the product of CO$_2$ hydrogenation aggravates the sintering of Ni species. Thus, the rational design of anti-sintering and water-resistant Ni-based catalysts is demanding to

[1]Institute of Industrial Catalysis, State Key Laboratory of Green Chemistry Synthesis Technology, College of Chemical Engineering, Zhejiang University of Technology, Hangzhou, Zhejiang 310014, China. [2]Zhejiang Carbon Neutral Innovation Institute & Zhejiang International Cooperation Base for Science and Technology on Carbon Emission Reduction and Monitoring, Zhejiang University of Technology, Hangzhou 310014, China. [3]Key Laboratory of Biomass Chemical Engineering of Ministry of Education, College of Chemical and Biological Engineering, Zhejiang University Hangzhou 310027, China. [4]These authors contributed equally: Xin Tang, Chuqiao Song. ✉e-mail: yaosiyu@zju.edu.cn; linll@zjut.edu.cn

overcome the stability challenges in the $CO_2$ hydrogenation to methane[15].

Structured metal materials like Ni-foam and Cu-foam etc. with high heat conductivity, rich channels and mechanical robustness provide opportunities to eliminate the undesirable generation of local hotspots[16–20]. However, the applications of metal-structured catalysts are limited due to the unfavorable catalytic functionalization of active sites (Ni/oxide) and poor adherence of metal oxides on the surface of metal foam skeleton (Ni/oxide/Ni-foam)[21–25]. Regarding the existing challenges and inspired by our previous studies on the inverse catalysts with improved $CO_2$ hydrogenation performances, we propose the construction of nano-oxide/Ni inverse structure on Ni-foam as the active site for $CO_2$ methanation in order to exploit the advantages of structured and inverse catalysts[26–32]. Particularly, by growing a closely contact layer of nickel hydroxide on the Ni-foam substrate via an etching process as the attaching sites of nano-oxides, a fine and uniform dispersion of oxide/NiO nano-composites over Ni foam with high density and strong structure robustness can be obtained as the precursor of oxide/Ni inverse structure[33–35]. The inverse oxide/Ni active species functionalized Ni foam structured catalyst will simultaneously enhance the $CO_2$ hydrogenation activities and realized remarkable stability.

In this work, we report a Ni-foam supported inverse $CeAlO_x$/Ni species ($CeAlO_x$/Ni/Ni-foam) as an efficient structured catalyst for $CO_2$ hydrogenation towards methane. The inverse $CeAlO_x$/Ni/Ni-foam catalyst presents significantly improved methane productivity at low temperature and exhibits superior thermal stability, and its activity remains virtually unchanged after seven cycles of heating-cooling treatment (25–600 °C) and 200 h time-on-stream (at 240 °C) without significant sintering or carbon deposition. The structured catalyst also shows excellent water resistance, and the $CO_2$ methanation activity can be reversibly recovered after the removal of excessive steam. Besides the excellent stability, the structured catalyst also realizes a $CO_2$ conversion above 80% at 240 °C with a $CH_4$ selectivity over 98.6% at GHSV of 80,000 h$^{-1}$, 14 times higher than the conventional Ni/oxide references. This design and fabrication of the structured catalyst with inverse species as active sites provide a general strategy and a promising platform to construct high-performance and durable catalysts for $CO_2$ hydrogenation reaction to methane.

## Result

### Structural characterization of catalysts

The $Ni(OH)_2$ overlayer-covered Ni-foam is prepared using a urea hydrothermal etching method. The following modification of the $Ni(OH)_2$ layer with Ce and Al oxides is realized by hydrothermal method followed by calcination at 400 °C (Fig. 1a). The prepared catalyst is labeled as $CeAlO_x$/NiO/Ni-foam, and the loadings of Al and Ce are 2.5 wt.% and 2.4 wt.% (about 10.4 wt.% and 11.2 wt.% respective to NiO overlayer, determined by inductively coupled plasma-optical emission spectrometer (ICP-OES)). Other reference catalysts including the $Al_2O_3$/NiO/Ni-foam and NiO/Ni-foam are prepared with the same procedure. Before performance evaluation, all catalysts are pre-reduced in 20% $H_2$ at 450 °C for 3 h to convert the NiO substrate into metallic Ni to generate the inverse oxide/Ni composites on Ni foam skeleton (labeled as $CeAlO_x$/Ni/Ni-foam, $Al_2O_3$/Ni/Ni-foam and Ni/Ni-foam). Ni supported on the $Al_2O_3$ and $CeAlO_x$ oxide supports are prepared by the precipitation method (Ni loading is controlled at 13 wt%) to compare with the inverse oxide/Ni composite catalysts, which helps to understand the importance of Ce doping.

X-ray diffraction (XRD) (Supplementary Fig. 1) patterns of the $CeAlO_x$/NiO/Ni-foam, $Al_2O_3$/NiO/Ni-foam and NiO/Ni-foam show intense diffraction peaks corresponding to NiO and Ni-foam substrate. After modification with Ce and Al oxides, broad peaks at 20°-25° appear in the $CeAlO_x$/NiO/Ni-foam and $Al_2O_3$/NiO/Ni-foam, which suggests fine dispersion of Ce and Al oxide species on the substrate

due to the anchoring of the $Ni(OH)_2$ overlayer[36,37]. All catalysts exhibit type-IV isotherms with type-H3 hysteresis loops, indicating the presence of mesopores (Supplementary Fig. 2) and the Barrett-Joyner-Halenda apertures in the structured catalysts[38]. The average pore sizes of $CeAlO_x$/NiO/Ni-foam, $Al_2O_3$/NiO/Ni-foam, and NiO/Ni-foam catalysts are ~6 nm, 5.5 nm, and 4.3 nm (Supplementary Table 1).

The morphology and microstructure of $CeAlO_x$/NiO/Ni-foam are further observed using electron microscopic methods. The scanning electron microscopy (SEM) images of $CeAlO_x$/NiO/Ni-foam preserves monolith geometry and rich 3-dimensional cross-connected pore structure after the modification and thermal treatments (Supplementary Fig. 3). Th $CeAlO_x$/NiO composite displays a honeycomb-like nanoflake appearance on the skeleton of Ni foam with an average thickness of 4 μm (Fig. 1b and Supplementary Fig. 3). The adherence of $CeAlO_x$/NiO on Ni foam is sufficiently strong to bare the vigorous ultrasonic treatment (Supplementary Fig. 5), which highlights the effectiveness of the NiO overlayer in anchoring the fine oxide species. Transmission electron microscopy (TEM) images of $CeAlO_x$/NiO sample scraped from the structured catalysts suggest that NiO nano-particles (distribution centered at ~5.4 nm) are deposited on the exterior surface of Ni-foam (Fig. 1c and Supplementary Fig. 4). High angle dark field scanning transmission electron microscopy (HAADF-STEM) and energy-dispersive X-ray (EDS) element mapping images further confirm the uniform dispersion of Ce and Al over NiO particles in the nano-composite (Fig. 1d, e). Atomic-level image of region 1# from Fig. 1d shows the lattice fringes of 0.265 nm (correspond to $CeAlO_3$(110), Supplementary Fig. 6), demonstrating the formation of $CeAlO_x$ mixed oxide and loaded on the NiO support (Fig. 1f). After reduction, an inverse interface composed with $CeAlO_x$ oxide particles on Ni support will be formed. Raman spectroscopy is performed to investigate the metal-O vibration of different Ni-foam structured catalyst (Fig. 1g). The peak at 540 - 650 cm$^{-1}$ is confirmed to the contribution of Ni-O based on the comparison of passivated $MO_x$/NiO/Ni-foam and reduced $MO_x$/Ni/Ni-foam catalyst, as the Ni-O vibration peak disappears completely (Supplementary Fig. 7) due to the fully reduction of NiO to metallic Ni[39]. The redshift of Ni-O vibration peaks in the $Al_2O_3$/NiO/Ni-foam (580 cm$^{-1}$) compares to that of NiO/Ni-foam (540 cm$^{-1}$), which is probably the effect of the formation of Al-O-Ni coordination. Then, a larger red shift of Ni-O vibration appears when Ce is introduced to the $Al_2O_3$/NiO/Ni-foam inverse catalyst, and no Ce-O vibration is emerged, suggesting the formation of $CeAlO_x$ mixed oxide which affects the Ni-O vibration[40].

The chemical state of the catalyst surface is further explored by in situ X-ray photoelectron spectroscopy (XPS, peak fitting results in Supplementary Fig. 8 and Supplementary Table 2). From Ni 2$p$ XPS spectra (Fig. 1h), it is confirmed that the surface of calcined $CeAlO_x$/NiO/Ni-foam mainly corresponds to $Ni^{2+}$ species (>76%), which converts into metallic $Ni^0$ after reduction[41,42]. The Ce 3$d$ (Fig. 1h) spectra show that over 60% surface Ce atoms become to $Ce^{3+}$ species after reduction, which could introduce abundant oxygen vacancies in the inverse composite. Meanwhile, the 0.8 eV negative shift of the Al 2$p$ XPS peak of the $CeAlO_x$/Ni/Ni-foam sample compared with $Al_2O_3$/Ni/Ni-foam sample demonstrates the formation of $CeAlO_x$ mixed metal oxides in the catalyst (Fig. 1i)[39,43]. The $O_{surface}/(O_{surface}+O_{lattice})$ ratio of $CeAlO_x$/Ni/Ni-foam catalyst reaches ~30% (based on the O 1$s$ region XPS spectra in Supplementary Fig. 6), which is in good agreement with the $CeAlO_x$ mixed metal oxides contains higher density of oxygen vacancies based on $O_2$-pulse chemisorption results (Supplementary Fig. 9).

### Catalytic performance of the structured catalysts

The catalytic performances of the structured catalysts for $CH_4$ synthesis from $CO_2$ hydrogenation are evaluated between 160–300 °C using a gas feed of $CO_2/H_2/N_2$ = 18/72/10 under atmospheric pressure and a gas hourly space velocity (GHSV) of 10,000 h$^{-1}$. The activities of

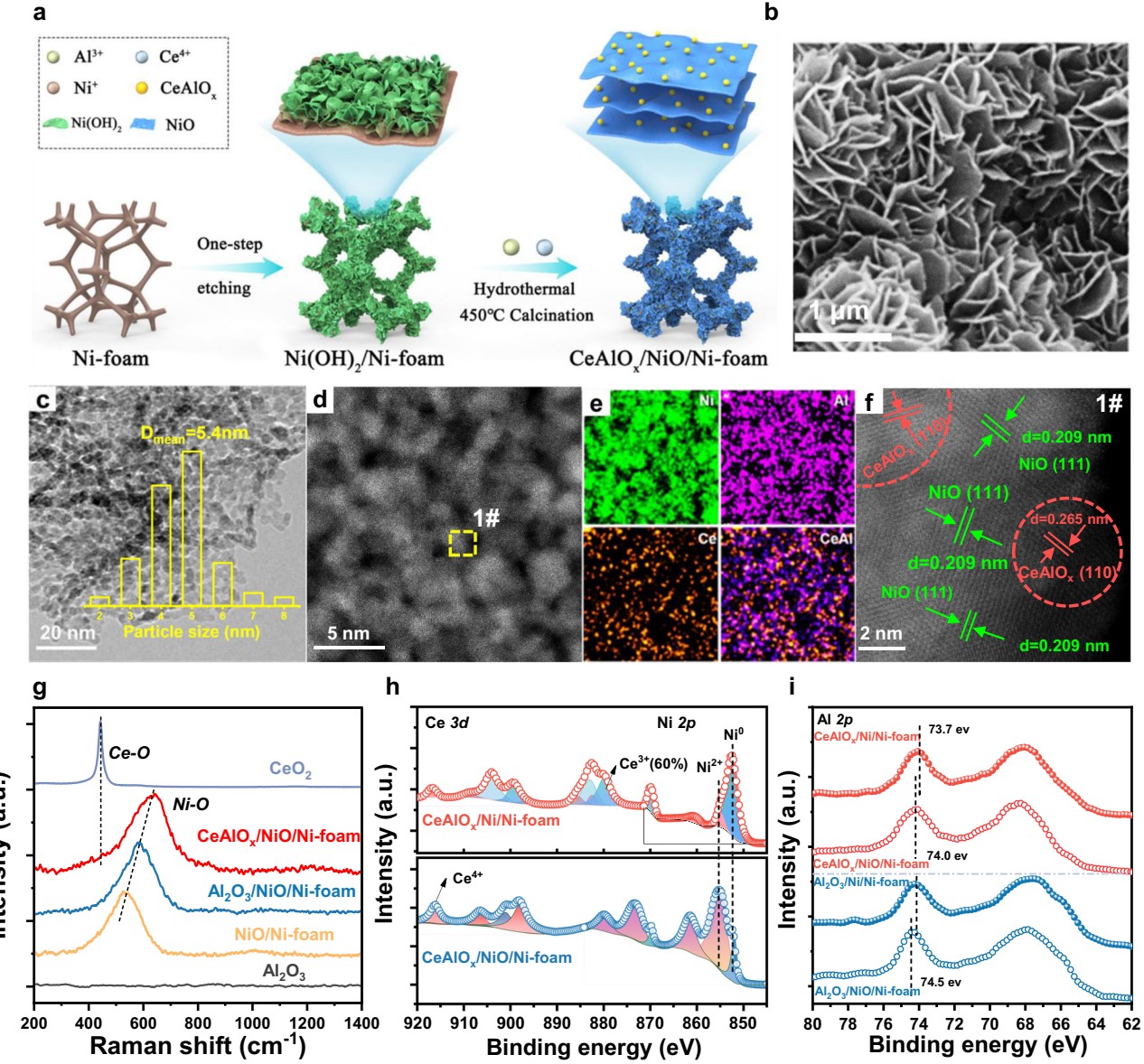

**Fig. 1 | Catalyst preparation strategy and structure characterization.**
**a** Schematic diagram of the synthesis of CeAlOx/NiO/Ni-foam catalyst; **b** SEM image of CeAlOx/Ni(OH)2/Ni-foam catalyst; **c** TEM image of CeAlOx/NiO catalyst scraped from the Ni-foam substrate (inset is the particle size distribution histogram); **d** Aberration-corrected HAADF-STEM image of scraped CeAlOx/NiO catalyst; **e** EDS elemental maps of scraped CeAlOx/NiO catalyst, showing the distribution of Ni, Ce and Al; **f** High-resolution HAADF-STEM image of the 1# area in **d**; **g** Raman spectra of the NiO/Ni-foam, Al2O3/NiO/Ni-foam, CeAlOx/NiO/Ni-foam, CeO2 and Al2O3 catalysts; **h** In situ XPS of Ce 3d and Ni 2p of CeAlOx/NiO/Ni-foam and CeAlOx/Ni/Ni-foam catalysts; **i** In situ XPS of Al 2p of CeAlOx/NiO/Ni-foam, CeAlOx/Ni/Ni-foam, Al2O3/NiO/Ni-foam and Al2O3/Ni/Ni-foam catalysts.

MOx/Ni/Ni-foam (M = Y, Zr, Al, Ce, and Mg) catalysts and the Ni/Ni-foam catalyst in $CO_2$ methanation reaction are showed in Fig. 2a, b and Supplementary Fig. 10. Almost no $CO_2$ conversion is observed over the Ni/Ni-foam and Ni-foam substrates (below 250 °C). In comparison, 40–80% $CO_2$ conversion are obtained at 250 °C on the MOx/Ni/Ni-foam catalysts, suggesting the importance of oxide modification in promoting the $CO_2$ methanation activity. Furthermore, it is found the formation of Ce-Al mixed oxide phase (CeAlOx/Ni/Ni-foam catalyst) doubles the $CO_2$ conversion at 200 °C compared with Al2O3/Ni/Ni-foam (Fig. 2a, b and Supplementary Fig. 11) and Ce/Al/Ni=1/5/30 is determined as the optimal composition. In the performance evaluation, CeAlOx/Ni/Ni-foam catalyst achieves ~90% $CO_2$ conversion and $CH_4$ selectivity of >99.9% at 240 °C, which far exceeds the conventional oxide-supported Ni catalysts. The space-time yields (STY) of $CH_4$ of CeAlOx/Ni/Ni-Foam and corresponding Ni/CeAlOx catalyst in kinetic region ($CO_2$ conversion<15%[44], Supplementary Fig. 12 and Supplementary Table 4) show that the $CH_4$-STY of CeAlOx/Ni/Ni-foam catalyst is 65.3 mmol$_{CH_4}$/mL$_{foam}$/h, which is 15 times higher than that of Ni/CeAlOx catalyst.

For the comparison of $CO_2$ and $H_2$ reaction order, diluting $CO_2$ reaction gas was applied to ensure that $CO_2$ is converted in the kinetic region and the effect of hotspots is eliminated. Kinetic analysis of the $CO_2$ methanation catalysts shows the apparent $H_2$ and $CO_2$ reaction orders of CeAlOx/Ni/Ni-foam are 0.34 and 0.21, and those of Al2O3/Ni/Ni-foam are 0.36 and 0.24. In comparison, the reaction orders of conventional Ni/CeAlOx and Ni/Al2O3 are 0.81/0.02 and 0.82/0.04 (Fig. 2c and Supplementary Table 5). The change of the apparent kinetic orders of $H_2$ and $CO_2$ suggests that the $CO_2$ coverage decreases and $H_2$ surface coverage is intensified over the MOx ensembles of CeAlOx/Ni/Ni-foam and Al2O3/Ni/Ni-foam according

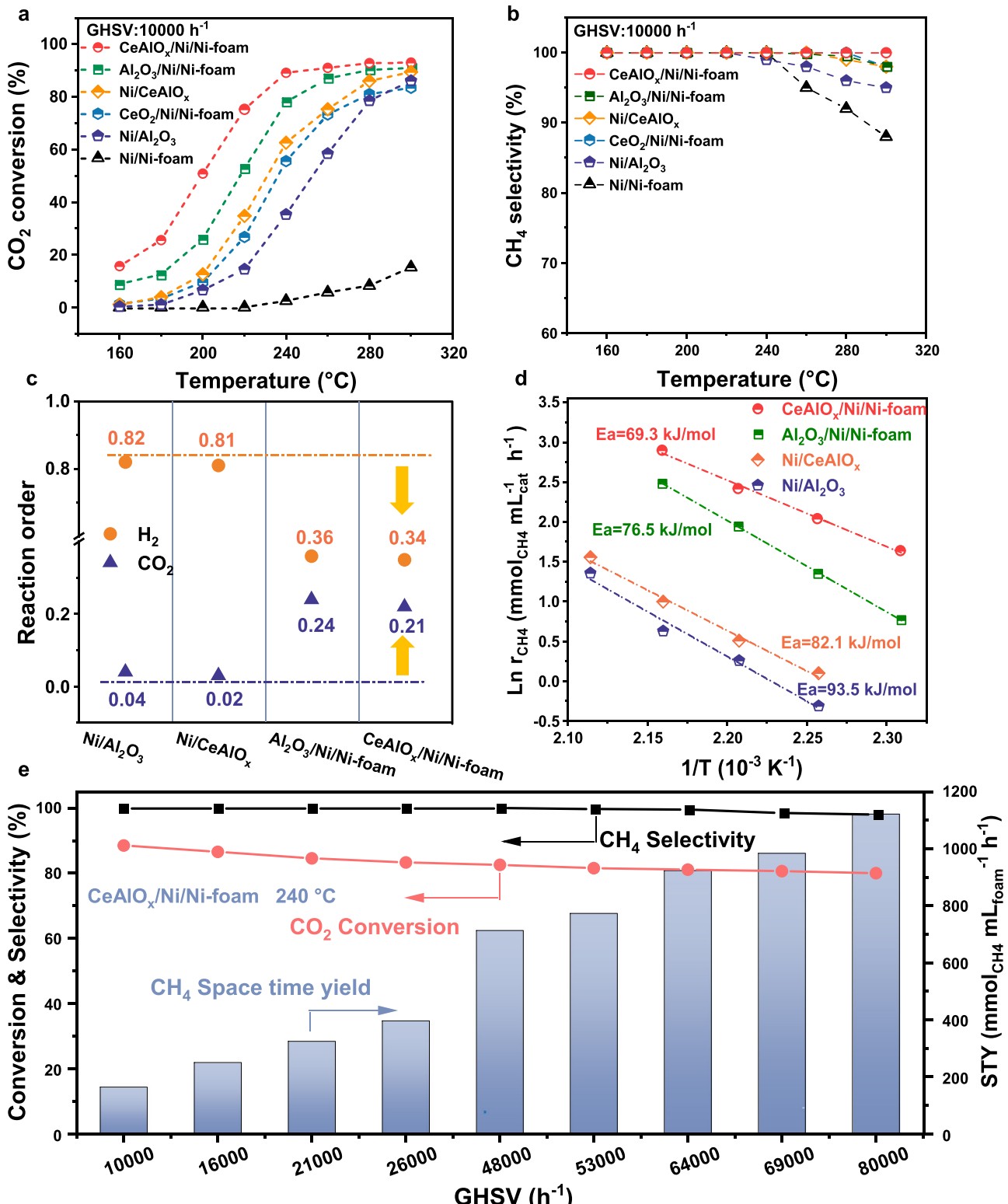

**Fig. 2 | The catalytic performance of CeAlO_x/Ni/Ni-foam catalyst.** Temperature-dependent **a** $CO_2$ conversion and **b** $CH_4$ selectivity of the CeAlO_x/Ni/Ni-foam, Al_2O_3/Ni-foam, CeO_2/Ni/Ni-foam, Ni/Al_2O_3, and Ni/Ni-foam catalysts (reaction conditions: GHSV = 10,000 h^-1, 160–300 °C $CO_2$:$H_2$:$N_2$ = 18:72:10, P = 0.1 MPa); **c** Reaction orders with respect to $H_2$ and $CO_2$ for methane formation; **d** $CH_4$ based apparent activation energy ($E_a$) of CeAlO_x/Ni/Ni-foam, Al_2O_3/Ni-foam, Ni/Al_2O_3 and Ni/CeAlO_x catalysts; **e** GHSV-dependent activities of CeAlO_x/Ni/Ni-foam catalyst at 240 °C.

to the Langmuir-Hinshelwood mechanism, which could significantly promote the surface reaction. By varying the GHSV for different catalysts, it is ensured that all the $CO_2$ conversion used to calculate $E_a$ are below 6% (Supplementary Fig. 13). The $CH_4$ base $E_a$ of CeAlO_x/Ni/Ni-foam is determined as 61.3 kJ/mol, lower than Al_2O_3/Ni/Ni-foam (76.5 kJ/mol) and much lower than that of conventional Ni/CeAlO_x (82.1 kJ/mol) and Ni/Al_2O_3 (93.5 kJ/mol), confirming the immense contribution of CeAlO_x/Ni inverse structure on promotion of reaction

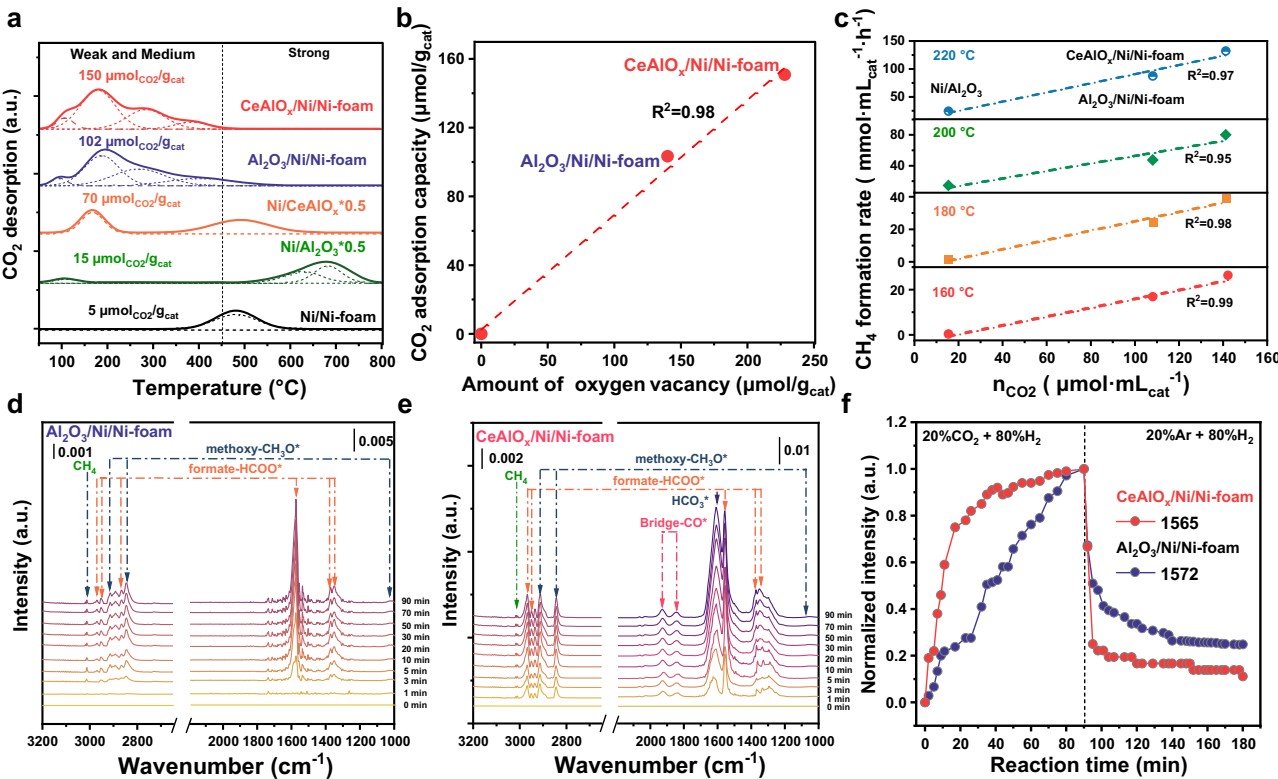

**Fig. 3 | Investigation on the reaction mechanism of CeAlO$_x$/Ni/Ni-foam catalyst.** **a** The CO$_2$-TPD profiles of Ni-base catalysts; **b** relationship between CO$_2$ capture capacity and amount of oxygen vacancies on CeAlO$_x$/Ni/Ni-foam and Al$_2$O$_3$/Ni/Ni-foam catalysts; **c** the correlation of the STY methanol and the amount of adsorbed CO$_2$ at 50–400 °C; **d**, **e** DRIFTs results of t the Al$_2$O$_3$/Ni/Ni-foam catalyst and CeAlO$_x$/Ni/Ni-foam catalyst in the stream of CO$_2$/H$_2$ mixture under 0.1 MPa respectively at 180 °C; **f** normalized intensities of the typical formate surface species as a function of reaction time (-1565 cm$^{-1}$ for CeAlO$_x$/Ni/Ni-foam; -1572 cm$^{-1}$ for Al$_2$O$_3$/Ni/Ni-foam).

kinetics in methane synthesis from CO$_2$ hydrogenation reaction (Fig. 2d).

The STY of CH$_4$ as a function of GHSV at 240 °C is further evaluated (Fig. 2e), and it is found that the CO$_2$ conversion of CeAlO$_x$/Ni/Ni-foam remains above 80% when the GHSV increases to 80,000 h$^{-1}$. The corresponding STY of methane reaches 1109 mmol$_{CH4}$/mL$_{foam}$/h (4450 mmol/g$_{cat}$/h with respect to the mass of CeAlO$_x$/Ni ensemble), which is more competitive than the state-of-the-art supported Ru and Ni catalysts for the low-temperature CO$_2$ methanation (Supplementary Table 6).

To understand the excellent catalytic performance of the CeAlO$_x$/Ni/Ni-foam structured catalyst, a number of characterizations are performed to identify the active sites. CO$_2$ temperature program desorption profiles (Fig. 3a) show the amount of CO$_2$ adsorbed at weak and medium alkaline sites are 150 and 102 μmol/g$_{cat}$ (Supplementary Table 3). It can be seen that the capacity of weak- and medium-adsorbed CO$_2$ display a near linear correlation with the density of oxygen vacancies ($R^2 = 0.98$) (Fig. 3b). As the weak- and medium-adsorbed CO$_2$ are determined to show a linear relationship with the intrinsic productivity of CH$_4$ at 160, 180, 200, and 220 °C (Fig. 3c), it can be confirmed that the oxygen vacancies at the inverse oxide-metal interface are probably the sites for CO$_2$ activation at low temperature, which accounts for the activity of CO$_2$ methanation.

In situ diffuse reflectance Fourier transform infrared spectroscopy (DRIFTS) studies further elucidate that the structure of the MO$_x$/Ni/Ni-foam composites affect the types and conversion rate of surface intermediates (Fig. 3d, e and Supplementary Fig. 14). Under the reaction atmosphere (CO$_2$ + 4H$_2$), bridged CO* (1833 and 1930 cm$^{-1}$), formate (2970, 1563, 1380 cm$^{-1}$) and methoxy (2845, 2926 cm$^{-1}$) species are observed on CeAlO$_x$/Ni/Ni-foam catalyst[45]. In contrast, only

formate and methoxy species are observed on Al$_2$O$_3$/Ni/Ni-foam catalyst (Fig. 3d). In addition, when CO$_2$ is removed from the feed after steady state is reached, CO* and formate species on the CeAlO$_x$/Ni/Ni-foam catalyst are rapidly consumed together with the formation of methane, and the consumption of formate species and formation of methane is also observed on the Al$_2$O$_3$/Ni/Ni-foam (Fig. 3d and Supplementary Fig. 14), which indicates that both formate and CO* are important intermediates on the CeAlO$_x$/Ni/Ni-foam catalyst, while methanation on the Al$_2$O$_3$/Ni catalyst mainly follows the formate pathway. Therefore, these two possible reaction pathways synergistically promote the lower temperature methanation on the CeAlO$_x$/Ni/Ni-foam catalyst.

## Mechanism studies

The reaction stability is probably one of the most important indicators for a practical catalyst, especially for the CO$_2$ methanation catalyst, which faces significant challenges of sintering and carbon deposition[46]. To investigate the thermal shock resistance of CeAlO$_x$/Ni/Ni-foam structure catalyst, a seven-cycle reciprocating heating-cooling test between 25 and 600 °C was performed (Fig. 4a). After each cycle, the CO$_2$ conversion and CH$_4$ selectivity of CeAlO$_x$/Ni/Ni-foam at 240 °C can be restored (Fig. 4a). In contrast, the conventional Ni/CeAlO$_x$ shows a rapid deactivation after only one heating-cooling cycle (Fig. 4b), which is probably due to the agglomeration of Ni NPs (see XRD patterns of the fresh and spent catalysts in Fig. 4c). This phenomenon implies that the interaction between the oxide and Ni substrate effectively inhibit the migration of Ni species and thereby prevent the undesirable sintering[47,48]. Additionally, the temperature program oxidation (TPO) experiment of the spent CeAlO$_x$/Ni/Ni-foam and Ni/CeAlO$_x$ catalysts also confirms coke formed on

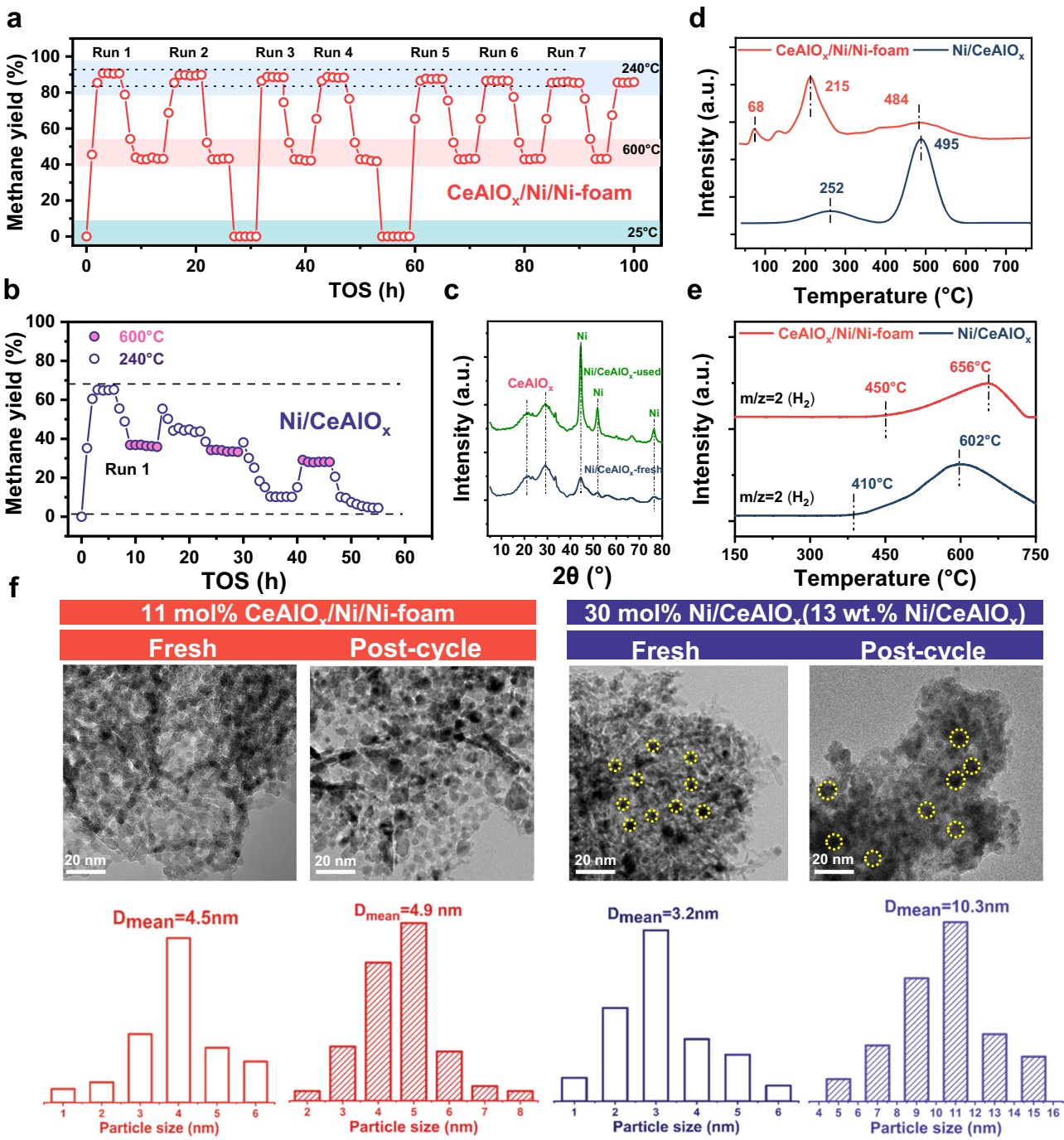

**Fig. 4 | Thermal shock resistance of CeAlOx/Ni/Ni-foam catalyst.** CH4 yield of **a** CeAlOx/Ni/Ni-foam and **b** Ni/CeAlOx catalysts during heating-cooling treatment (reaction conditions: GHSV = 10,000 h⁻¹, CO2:H2:N2 = 18:72:10, *P* = 0.1 MPa); **c** XRD spectra of Ni/CeAlOx catalyst before and after cyclic reaction; **d** TPO results of CeAlOx/Ni/Ni-foam and Ni/CeAlOx catalysts after heating-cooling cycle tests; **e** TPSR results of methane on Ni/CeAlOx and CeAlOx/Ni/Ni-foam. Reaction conditions: 10 vol% CH4/Ar, GHSV = 15,000 h⁻¹; **f** STEM images of CeAlOx/Ni/Ni-foam and Ni/CeAlOx catalysts before and after heating-cooling cycle tests.

CeAlOx/Ni/Ni-foam after seven cycles is mainly amorphous carbon which can be oxidized around 215 °C. While large amount of partial crystalized carbon is generated on Ni/CeAlOx after three cycles (mainly oxidized at 400-550 °C), demonstrating CeAlOx/Ni/Ni-foam structured catalyst is able to inhibit the formation of coke in CO2 methanation reaction (Fig. 4d). The coking resistance mechanism of the CeAlOx/Ni/Ni-foam can be illustrated by the CH4 temperature program surface reaction experiment (Fig. 4e), which indicates the decomposition of CH4 to H2 and carbon on the CeAlOx/Ni/Ni-foam is about 50 °C higher than the conventional Ni/CeAlOx catalyst. Moreover, the size of the

scraped CeAlOx/Ni inverse species before and after cycling experiments maintains a fine dispersion without agglomeration (4.5 nm to 4.9 nm) (Fig. 4f). On the contrary, the Ni NPs over Ni/CeAlOx sinters from 3.2 nm to 10.3 nm after four heating-cooling cycle, which explains the reason for the deactivation of conventional Ni/oxide catalysts.

The excellent thermal stability of structural catalysts compared with the conventional supported catalyst is also probably due to the improved heat and mass transport efficiency. The temperature rise of the catalyst bed is limited below 3 °C in a wide range of reaction temperature and CO2 conversion on the CeAlOx/Ni/Ni-foam,

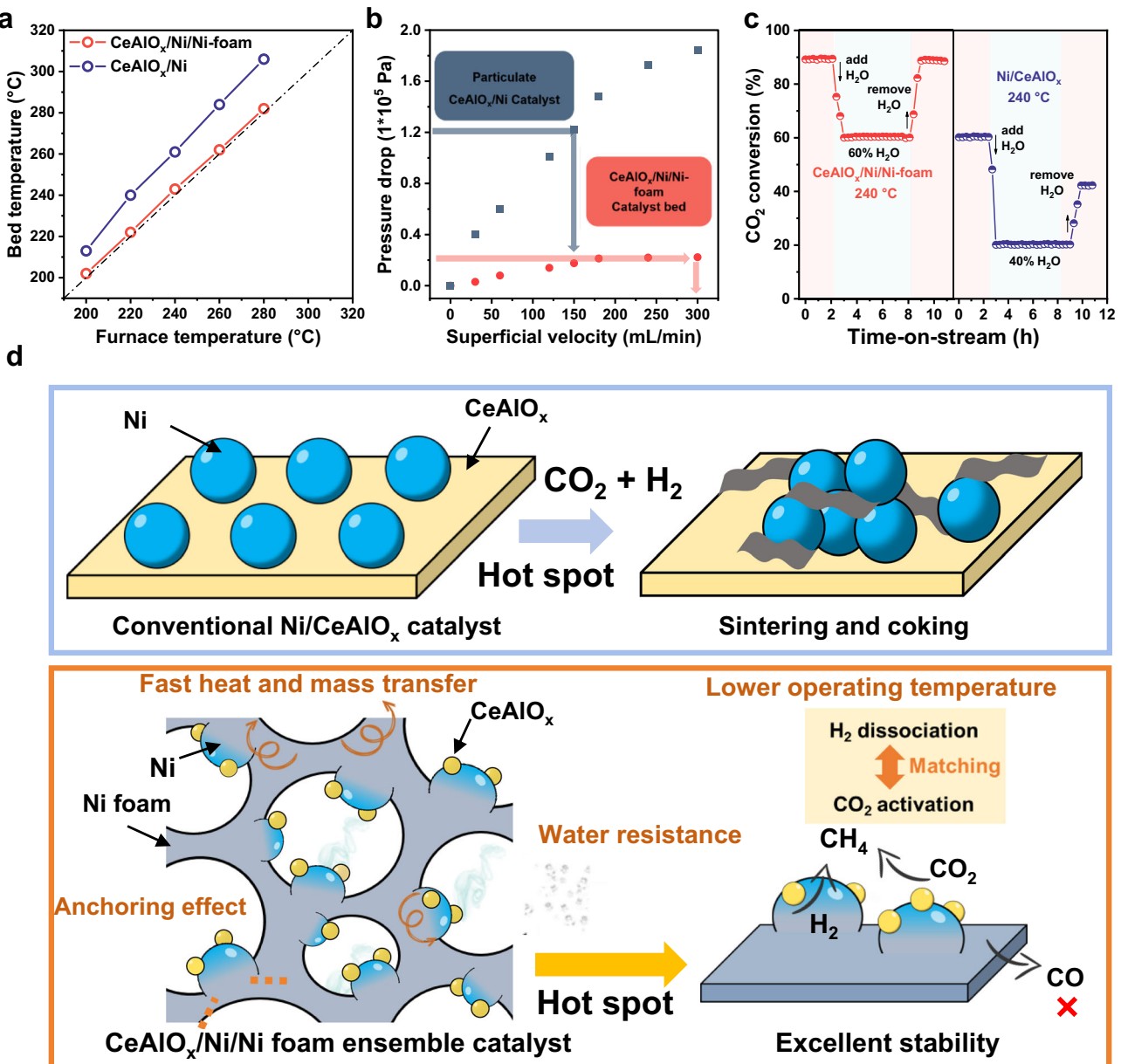

**Fig. 5 | Investigation of the thermal stability of structural catalyst. a** The comparison of temperature-rising for the Ni-foam-structured CeAlO$_x$/Ni catalyst and CeAlO$_x$/Ni catalyst; **b** pressure drop against N$_2$ gas superficial velocity, CeAlO$_x$/Ni/Ni-foam (100 PPI), CeAlO$_x$/Ni (60–80 meshes); **c** water resistance test of CeAlO$_x$/ Ni/Ni-foam catalyst (reaction conditions: 240 °C, GHSV = 10,000 h$^{-1}$, CO$_2$:H$_2$:N$_2$ = 18:72:10, $P$ = 0.1 MPa); **d** schematic representation of a Ni-foam skeleton constrained stabilized inverse nickel catalyst and a reference sample.

in contrast, without the Ni foam support, the temperature rise of CeAlO$_x$/Ni powder catalyst bed is above 20 °C (Fig. 5a), indicating the diminish of the localized hotspots can be highly due to the construction of structured catalysts. The pressure drops comparison of the CeAlO$_x$/Ni /Ni-foam structure catalyst and CeAlO$_x$/Ni powder catalyst suggest that the pressure drop of the structured catalyst is only 1/9 of the powder catalysts (0.2 × 10$^5$ Pa at the superficial velocity of 300 mL/ min, Fig. 5b). This enhanaced mass transfer efficiency probably also contributes to the hotspot elimination of the nickel foam-based catalyst. Additionally, since steam is one of the main products during methanation reaction, an additional amount of steam is introduced (60 vol% H$_2$O) at 240 °C to investigate the water resistent property (Fig. 5c). Ni-structured catalyst loses ~1/3 of its under the reaction condition of 60 vol% H$_2$O, but the catalytic activity can be totally recovered after the removal of steam[49,50]. In contrast, the activity of powder catalyst is lost more than 2/3, and only 70% catalytic activity

can be recovered after removing steam. The much better water resistence of CeAlO$_x$/Ni/Ni-foam structure catalyst can also be attributed to the porous structure that accelerates the diffusion of steam in the reaction.

Based on the performance and cylic stability tests for CO$_2$ hydrogenation to methane, the structured catalyst with Ni foam skeleton and well-designed inverse CeAlO$_x$/Ni species as active sites is demonstrated to display superior activity, stability and strong adaptability to unsteady operation condition and condensation compared with conventional oxide supported Ni-based catalysts (Fig. 5d). The high thermal conductivity of metal framework and the rich diffusion channels in the structured support successfully eliminate the local hotspots and prevent the accumulation of water surrounding the active sites, which benefits the thermal stability, coke elimination and water resistance. The inverse species which reduces the CO$_2$ coverage and accelerates the reduction of CO$_2$ and intermediates,

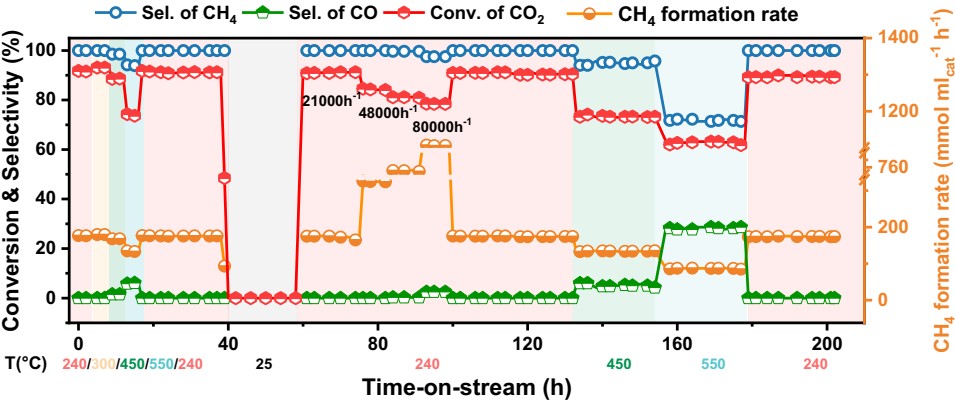

**Fig. 6 | Resistance to fluctuating conditions of CeAlO$_x$/Ni/Ni-foam catalysts.** Long-term stability test on CeAlO$_x$/Ni/Ni foam catalyst. Reaction conditions: 240–550 °C, GHSV = 10,000 h$^{-1}$, CO$_2$:H$_2$:N$_2$ = 18:72:10, $P$ = 0.1 MPa.

not only enhances the activity but also reduces the coke formation due to the successful suppress of CH$_4$ decomposition side reactions. Meanwhile, the finely dispersed metal oxide species on inverse MO$_x$/Ni composites also enhances the anti-sintering ability of CeAlO$_x$/Ni/Ni-foam catalyst and enhances the structure robustness of active species.

As CO$_2$ methanation is a potential reaction to integrate with unstable and discontinuous hydrogen production from renewable energy, the catalyst developed for the process need to be adaptive to unsteady operation condition and potential steam condensation[51]. Therefore, an unsteady operation condition with waving temperature and space velocity is set to simulate application scenarios and evaluate the stability of CeAlO$_x$/Ni/Ni-foam structured catalyst (Fig. 6). No sign of deactivation of catalyst is observed after 200 h time on stream, suggesting the application perspective of CeAlO$_x$/Ni/Ni-foam structured catalyst in hydrogen to gas processes.

## Discussion

In summary, a highly active, selective and thermally stable structured catalyst with inverse CeAlO$_x$/Ni ensemble active sites loaded on Ni-foam is successfully prepared and applied for the CO$_2$ hydrogenation to methane reaction. We demonstrate that the formation of CeAlO$_x$ mixed oxide on Ni enhances the oxygen vacancies for CO$_2$ activation and simultaneously modulates the surface coverage of CO$_2$ and hydrogen, which not only promotes the methanation activity by 14 times but also suppresses the decomposition of CH$_4$. Powered with the remarkable heat and mass transport efficiency of 3D Ni-foam and the excellent anchoring effect of Ni(OH)$_2$ overlayer prepared by the urea-etching method, the local hotspots are eliminated, and the structure of inverse ensemble is demonstrated to be intact after long-term unsteady operation or treated with steam-rich atmosphere, which overcomes the inherent stability challenges existed in the conventional supported-based catalysts. The development of the CeAlO$_x$/Ni/Ni-foam structured catalyst provides rational strategy to construct highly stable and affordable practical catalysts for CO$_2$ methanation reaction.

## Methods
### Materials
Analytical grade chemicals including the sodium carbonate (Na$_2$CO$_3$, 99 wt% purity), sodium hydroxide (NaOH, 99 wt% purity), nickelous nitrate hexahydrate (Ni(NO$_3$)$_2$·6H$_2$O, 98 wt% purity), cerium nitrate hexahydrate (Ce(NO$_3$)$_2$·6H$_2$O, 99 wt% purity) and aluminum nitrate nonahydrate (Al(NO$_3$)$_3$·9H$_2$O, 99 wt% purity) was purchased from Sinopharm Chemical Reagent Co., Ltd. The Ni-foam felt was purchased from Suzhou Taili Material Co. All chemicals were used as received without any further purification.

### Catalyst synthesis
**Preparation of CeAlO$_x$/Ni/Ni-foam catalyst.** The Ni(OH)$_2$/Ni foam substrate is prepared first. In a typical synthesis procedure, circular Ni foam thin slices (1 g, diameter 6 mm, thickness 1.0 mm, porosity 110 PPI) are cut from Ni foam plates and sonicated in acetone for 20 min to remove surface residual organic impurities. These circular slices are then immersed in a 0.1 M HCl solution at room temperature for an additional 20 min of sonication to remove the surface nickel oxide from the Ni foam, followed by thorough rinsing with deionized water. The cleaned Ni foam thin slices (0.4 g) are transferred to a stainless-steel autoclave lined with a 50 mL polytetrafluoroethylene (PTFE) container, which contains a 35 mL solution of urea (6.3 mmol). After hydrothermal treatment at 160 °C for 8 h, the Ni foam coated with deep green Ni(OH)$_2$ crystals is rinsed with deionized water and dried under vacuum at 80 °C for 12 h. A solution containing Al(NO$_3$)$_3$·9H$_2$O (0.875 mmol), Ce(NO$_3$)$_2$·6H$_2$O(0.218 mmol), and urea (6.5 mmol) is prepared (35 mL), and then the obtained solution is stirred for about 60 min. Subsequently, the resulting solution and Ni(OH)$_2$/Ni foam thin slices (0.4 g) are transferred to a Teflon-lined autoclave reactor (100 mL), subjected to hydrothermal treatment at 180 °C for 12 h. After cooling to room temperature, the sample is washed with ethanol and deionized water, dried under vacuum at 60 °C for 12 h, and finally calcined at 400 °C for 3 h to obtain the CeAlO$_x$/NiO/Ni foam catalyst.

**Preparation of Ni/CeAlO$_x$ catalyst.** The Ni/CeAlO$_x$ catalyst is prepared by a coprecipitation method. Briefly, a solution containing Al(NO$_3$)$_3$·9H$_2$O (5 mmol), Ce(NO$_3$)$_2$·6H$_2$O (1.25 mmol), and Ni(NO$_3$)$_2$·6H$_2$O (2.68 mmol) is prepared (100 mL), and then the aqueous metal precursor solutions are added dropwise to a precipitating solution of Na$_2$CO$_3$ and NaOH at vigorous stirring conditions. The resulting solution is stirred for 1 h, then maintain the pH to 10 by adding 3 M NaOH solution. After that, the precipitated mixture is aged at 65 °C in the reactor for 18 h to promote the crystallization of metals. Finally, the solid precipitate is filtered out be washed with ultrapure water many times to reduce the pH of the mixture to neutral. The obtained solid is dried at 110 °C overnight, and further calcined at 400 °C in air[52].

### Catalytic evaluation
The performance evaluation of CO$_2$ hydrogenation to methane is performed in an atmospheric fixed-bed reactor. The prepared catalyst sheets (0.15 g, diameter 6 mm) are loaded into a quartz tube (inner diameter = 6 mm and length = 60 cm) and put into the reactor. The catalyst is preprocessed in 20% H$_2$ at 450 °C for 3 h, cooled to the reaction temperature (160–300 °C), then the reaction gas (CO$_2$:H$_2$:N$_2$ = 18:72:10) is fed into the reactor. The actual temperature of the catalyst bed is measured using a thermocouple located at the

middle of the catalyst bed. Gas-phase products are analyzed using a gas chromatograph (GC-8860, Agilent) equipped with a thermal conductivity detector, Porapak Q and 5 A molecular sieve columns. The definitions of $CO_2$ conversion, $CH_4$ selectivity, carbon balance, and $CH_4$ STY are given by the following equations:

$$X(CO_2)\% = \frac{F * C_{in}(CO_2) - F * C_{out}(CO_2) * \frac{A_{in(N_2)}}{A_{out(N_2)}}}{F * C_{in}(CO_2)} \quad (1)$$

$$S(CH_4)\% = \frac{n(CH_4)}{\sum n(products)} \quad (2)$$

$$S(CO)\% = \frac{n(CO)}{\sum n(products)} \quad (3)$$

$$STY(CH_4)(mmol_{CH_4} \cdot ml_{cat}^{-1} \cdot h^{-1}) = \frac{n_{in}(CO_2) \cdot X(CO_2) \cdot S(CH_4) \cdot 16 \cdot 60}{22.4 \cdot V_{cat}} \quad (4)$$

where $F$ denotes the gas flow into the reactor, C denotes the concentration, $A$ denotes the gas chromatographic peak area, $V_{cat}$ denotes the volume of catalyst and $n$ denotes the amount of substance.

The Arrhenius plots were created at a high GHSV of 15,000–40,000 $h^{-1}$ to ensure that the concentration of carbon dioxide produced remained below 15%. This was achieved due to the insignificant influence of heat and mass transfer in this region. Additionally, differential mass-normalized reaction rates were calculated in the kinetic regime.

## Catalyst characterization

**Inductively coupled plasma-optical emission spectrometer.** The ICP-OES results are performed on Varian ICP-OES 720. Sample preparation: A certain number of samples are weighed into a PTFE container, added with 5 mL concentrated nitric acid, 3 mL HCl, 1 mL HF and 2 mL $H_2O_2$, sealed in a microwave digestion furnace, heated at 1200 W for 20 min to 130 °C, kept for 5 min, heated for 20 min to 180 °C, kept for 40 min, and cooled to room temperature. Test: The cooled solution is transferred to a 25 mL plastic volumetric bottle, and filled with deionized water. The dissolved solution is tested sequentially, and the diluted solution beyond the curve is tested again. Standard test solution: the standard solution is a national standard material, and the curve concentration points are 0, 0.5, 1.0, 2.0, 5.0 mg/L, respectively.

**X-ray diffraction.** XRD is used to determine the phase composition and estimate the particle size of the catalyst. The testing is conducted using a Cu-Kα excitation source with a scanning range of 2θ = 10° ~ 80°, a scanning speed of 20°/min, and a step size of 0.0167. The phase analysis is conducted by referring to the standard powder diffraction cards. The particle size of Ni is calculated using the Scherrer equation.

**Surface area measurement.** $N_2$ physical adsorption testing is conducted on the BSD-PS2 instrument. Prior to the testing, the sample is subjected to a vacuum degassing at 200 °C for 4 h, followed by $N_2$ adsorption-desorption testing under liquid nitrogen cooling (−196 °C) conditions. The determination of the specific surface area and distribution of pore sizes is accomplished through utilization of the Brunauer-Emmett-Teller (BET) method for calculation, in conjunction with analysis of the desorption curve using the Barrett-Joyner-Halenda (BJH) technique.

**$H_2$ temperature-programmed reduction ($H_2$-TPR).** $H_2$-TPR is conducted on the BELCAT-B instrument. A sample of 50 mg is weighed and pretreated in a flowing pure He gas (30 mL/min) for 1 h at 130 °C. After the sample is cooled to room temperature., a flow of $H_2$/Ar (10/90) gas (30 mL/min) is introduced. The temperature is then ramped from 50 °C to 700 °C with a heating rate of 10 °C/min for the temperature-programmed reduction process. The consumption of hydrogen is recorded by a thermal conductivity detector.

**$CO_2$ temperature-programmed desorption ($CO_2$-TPD).** $CO_2$-TPD is conducted on the Microtrac BEL Cat II instrument. The catalyst (50 mg) is pretreated at 450 °C for 180 min in 20% $H_2$ (heating rate of 5 °C/min), followed by cooling to 50 °C and purge with He for 30 min. Then, the catalyst is treated in $CO_2$/He (10/90) for 60 min, followed by a 40-min purge with He to remove unabsorbed and physically adsorbed $CO_2$. After the baseline has been stabilized, the temperature is gradually increased from room temperature to 800 °C at a heating rate of 10 °C/min in order to facilitate the desorption of $CO_2$.

**Temperature-programmed oxidation (TPO) of spent catalysts.** The catalysts, after stability test are exposed to 20% $O_2$/Ar (50 mL/min) at ambient temperature purge for 30 min, the fixed-bed reactor is heated to 700 °C with a rate of 10 °C/min and then held for 10 min. The CO and $CO_2$ are quantified by mass spectrum analyzer (DECRA), but $CO_2$ is the major product[53].

**Scanning electronic microscopy.** The samples were analyzed using a high-resolution field emission scanning electron microscope (FE-SEM, HITACHI Regulus 8100) operating at an acceleration voltage of 20 kV. Following that, the distribution of elements was determined utilizing EDX (Oxford Ultim Max 65).

**Transmission electron microscope.** TEM is conducted using a FEG-TEM instrument (Tecnai G2 F30 S-Twin) operating at 300 kV. The samples are sparsely dispersed in ethanol and subsequently deposited onto copper grids coated with amorphous carbon films, followed by desiccation for TEM observations[54].

**Scanning transmission electron microscope.** The Thermo Scientific Spectra 300 Double-Corrected Transmission Electron Microscope, equipped with a Gatan Imaging Filter, was utilized to conduct the STEM and EDX experiments. The point of scanning for elemental mapping within STEM-EDX was determined at 150×150. The pre-determined operating parameters necessitated the application of an acceleration voltage of 300 kV. To facilitate analysis and evaluation of the findings, the surface active phase CeAlOx/Ni from the reduced passivated nickel foam catalyst was extracted prior to TEM sample preparation for characterization.

**X-ray photoelectron spectroscopy.** X-ray Photoelectron Spectroscopy analysis is performed on a ThermoFischer ESCALAB 250Xi equipped with an in situ reactor. The specific parameters are as follows: excitation source using Al Kalpha radiation ($h\nu$ = 1486.6 eV); analysis chamber vacuum level of $8 \times 10^{-10}$ mbar; working voltage of 12.5 kV; filament current of 16 mA; and signal accumulation for ~10 cycles. The Passing Energy is set to 30 eV with a step size of 0.1 eV. The specific operational procedure is as follows: the catalyst sample, in the form of a disc, is placed inside the reactor chamber. It is pretreated for 1 h at a set temperature in an $H_2$/$N_2$ atmosphere (20 vol% $H_2$) with a flow rate of 20 mL/min. After cooling to room temperature, the sample is transferred to the measurement chamber without exposure to air. The measurement chamber is evacuated to a vacuum level below $8 \times 10^{-10}$ mbar before conducting the analysis. Charging correction of the binding energy is performed using C1$s$ (284.6 eV) as a reference.

**Raman spectroscopy analysis.** Raman spectra are obtained using the Renishaw In Via Reflex spectrometer with a 532 nm laser excitation source. The scanning range is set from 200 to 1800 $cm^{-1}$ with an accuracy of 2 $cm^{-1}$. The scan test is considered complete when

consistent results are obtained from at least three positions on each sample.

**Oxygen pulse titration (O$_2$-PT).** For the O$_2$ pulse experiments of NiO/Ni-foam, Al$_2$O$_3$/NiO/Ni-foam and CeAlO$_x$/NiO/Ni-foam catalysts are pretreated at 450 °C for 3 h under H$_2$ flow (20 vol% H$_2$/N$_2$, 40 mL/min), purged 10 min with He and heated to 500 °C. Then the 1% O$_2$ pulse experiments are repeated until the TCD peak intensity is equal.

$$O_{vacancy} = \frac{V(O_2) \cdot SF/22400}{\omega_{oxide} \cdot m_{cat}} \qquad (6)$$

where SF represents the stoichiometry factor, V(O$_2$) is the consumption of O$_2$ (deduct the Ni/Ni-foam consumption), $\omega_{oxide}$ is the oxide mass fraction (%), and $m_{cat}$ is the mass of the catalyst (g).

**Temperature-programmed surface reaction-mass spectrum.** The test procedure for CH$_4$ dissociation: 100 mg of sample, pretreat it at 450 °C for 3 h under 40 mL•min$^{-1}$ 20% H$_2$/Ar purge. Then cool down to room temperature (approximately 25 °C), and switch the 20% H$_2$/Ar to 25 mL•min$^{-1}$ 10% CH$_4$/Ar to record mass baseline. After the baseline is stable, the temperature is increased to 750 °C with a heating rate of 10 °C•min$^{-1}$, while the mass spectrum is recorded at the same time[55].

**In situ diffuse reflectance infrared flourier transform spectroscopy.** In situ DRIFTs measurements are performed by using an FTIR spectrometer (Bruker Vertex 80) equipped with a Harrick cell and a liquid nitrogen-cooled MCT detector, along with an RGA detector for the outlet gas analysis. The CeAlO$_x$/Ni/Ni-foam and Al$_2$O$_3$/Ni/Ni-foam catalysts are reduced in 10 mL min$^{-1}$ (H$_2$/Ar = 20/80) gas flow at 450 °C for 3 h, and then cooled down to 180 °C and purged with Ar for 30 min. The temperature of in situ DRIFTs is chosen to be 180 °C instead of 220 °C, in order to better observe the intermediate species at low activity. 1 min is averaged for each spectrum, which is recorded at a resolution of 4 cm$^{-1}$. Prior to each experiment, background is collected at Ar and 180 °C. Subsequently, the gas flow is changed to 80% H$_2$/20% CO$_2$ (10 mL min$^{-1}$, 0.1 MPa) at the same temperature, and the spectra are collected simultaneously. The transmittance is obtained by dividing the collected sample reflectance spectrum by the background spectrum, then spectrum is converted to Kubelka−Monk. After 90 min reaction in an 80% H$_2$/20% CO$_2$ atmosphere, the inlet is switched to 80% H$_2$/20% Ar (10 mL min$^{-1}$) at the same temperature. At the same time, DRIFTs spectra are recorded to monitor the change of intensity of different surface species for another 90 min.

## Data availability
The data that support the plots within this paper and another finding of this study are available from the corresponding author upon reasonable request. Source data are provided as a Source Data file. Source data are provided in this paper.

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

## Acknowledgements

This work is financially supported by the Zhejiang Provincial Natural Science Foundation of China (LR22B030003, LR21B030001 and LQ24B030016), Natural Science Foundation of China (22278367 and 22178302), National Key R&D Program of China (2022YFB4003100), China Postdoctoral Science Foundation Grant (2022M712817), Beijing National Laboratory for Molecular Sciences (BNLMS202011) and Research Fund of Department of Education of Zhejiang Province (Y202249632). We acknowledge the Electron Microscopy Center of Zhejiang University of Technology for the AC-TEM test.

## Author contributions

L.L., X.L., and S.Y. designed the study. X.T. performed most of the reactions. X.T. and Q.S. did the most data analysis. W.L., X.H., and H.-F.L. carried out the stability test. H.-B.L. carried out the DRIFTs analysis. Q.C. did STEM characterization. L.L., S.Y., Q.S., and X.T. wrote the paper. All authors performed certain experiments and discussed and revised the paper.

## Competing interests

The authors declare no competing interests.
