## [Peer Review File · Nature Communications]

Thermally Stable Ni Foam-Supported Inverse CeAlO_x/Ni Ensemble as an Active Structured Catalyst for CO₂ hydrogenation to MethaneREVIEWER COMMENTS

Reviewer #1 (Remarks to the Author):

In this manuscript the authors report a nickel-based catalyst with remarkable catalytic activity, stability, and outstanding water resistance for CO₂ methanation, which also shows resistance to carbon deposition. The origin of ultrahigh thermal stability is mentioned, but the discussion on catalytic mechanisms is insufficient. Therefore the work is still preliminary to publish as a qualified research. The authors need to improve their manuscript toward a high-level research.

(1) Some conclusions claimed by the authors are not supported by solid experimental data. This basically decreases the quality of this research. For example, the conditions which benefit the thermal stability, coke elimination and water resistance of the catalyst are explained by a model shown in Fig. 4c. This should be the critical novel point of this research, thus should be deeply discussed and revealed by probing the processes.

(2) In the conclusion part of this manuscript, the authors mentioned the adsorption/desorption behavior of CO₂ and hydrogen. However, the data about it is just a supplementary figure (Figure S7), which is not discussed in the main text.

(3) It is said in the manuscript that oxygen vacancy is the active site, but the working mechanism is not well explained.

Reviewer #2 (Remarks to the Author):

This manuscript presents a CeAlO_x/Ni/Ni-foam catalyst active for CO₂ methanation with exceptional stability against sintering and coking. To synthesize that material, the authors created a Ni(OH)₂ overlayer on Ni foam by urea-etching and deposited different metal oxides to create inverse structures. The conclusions are well supported by extensive characterization and thorough study of the stability of the materials. I very much enjoyed reading this elegant study, and I am confident it will be of interest to the readers of Nature Communications.

Below are some minor points that need attention:

- The manuscript and SI could benefit from further editions.
- The font in most figures is too small. Difficult to read.
- Page 4: "Ni supported on the Al₂O₃ and CeAlO_x oxide supports are prepared by the precipitation method (Ni loading is controlled at 13 wt%)." There is no mention to those catalysts before, so it might be confusing. I would recommend including a sentence explaining the motivation behind preparing those catalysts and further details about the synthesis.
- Page 4: "...Ni-foam substrate (marked with cubs)." There are more symbols in that figure, so "marked with cubs" could be omitted.
- Page 4: The term "hysterescence loops" is not correct. It should be "hysteresis loops."
- Page 4: Instead of "...suggest that NiO nanoparticles (~5.4 nm) are deposited..." it is recommended to use "...suggest that NiO nanoparticles (distribution centered at ~5.4 nm) are deposited..."
- Figure 2a: "Conversion of thermodynamic equilibrium" is too far from the dotted line, so the reader

might get confused.

- Figure 2: It is recommended to use different symbols as well, not only rely on different colors.
- Figure 2d: CeOx-Al₂O₃/Ni/Ni-foam shows a different nomenclature, better be consistent to avoid confusion.
- Page 5: "CeAlOx hetero-structure in the catalyst" is unclear. Is that a mixed metal oxide?
- Page 5: "CO₂ adsorption" would be more accurate than "CO₂ capture."
- Figure 3 legend: "after heating-cooling cycle tests" as there were more than one.
- Page 7: Missing information about how the authors ensured they worked in the kinetic region.
- Page 7: "inverse ensembles" might be difficult to understand. Suggestion: MOx ensembles.
- Page 7: "the coverage of CO₂ is reduced and H₂ surface coverage is intensified on the inverse ensembles" is difficult to read. Suggestion: "that the CO₂ coverage decreases and H₂ surface coverage is intensified over the MOx ensembles."
- References: Authors could include some references about the benefits of using inverse catalysts.
- SI, catalyst evaluation: Unclear what the authors mean with "a non-quartz tube." Is that a stainless-steel tube?
- SI, catalyst evaluation, equations: please define F, C, A, n, and V_{cat}.
- SI, catalyst characterization: missing details about ICP-AES.
- SI, STEM and EDX imaging: Using EDX and EDS in the text. Although it is the same, better use one to be consistent.
- Figure S1, legend: Also NiO/Ni-foam and Ni/Ni-foam.
- Figure S2: Used NF instead of "Ni-foam." Better be consistent to avoid confusion.
- Table S1: There is a typo. It should read as "CeAlOx/Ni/Ni-foam-used."
- Figure S9: It is recommended to use different symbols as well, not only rely on different colors.
- Figure S10: Lack of details about working in the kinetic region. Different GHSVs and temperatures.
- Table S4: Very diluted streams. It is assumed that is to get low CO₂ conversion (kinetic regime), but that's not mentioned in the main text.
- Table S5: RCH₄ seems to indicate productivity. The use of "R" might be confusing.

Reviewer #3 (Remarks to the Author):

This work deals with development of a new-type Ni-foam-structured inverse CeAlOx@Ni nanocomposite catalyst for application in the CO₂ methanation reaction. The results indicated that this approach worked effectively and efficiently to couple high activity/selectivity and promising stability with enhanced heat/mass transfer and high permeability. I recognized the relevance of the subject and the approach used by the authors is appropriate. In principle, I suggest accepting this draft for publication after MAJOR revision.

(1) Inverse structure of CeAlOx@Ni nanocomposite is not convincing, at least the evidences in the current manuscript is insufficient. High-quality images with clear inverse structure at interfaces between Ni particles and CeAlOx ensembles.

(2) The CeAlOx/Ni/Ni-foam catalyst is superior over the other MOx/Ni/Ni-foam catalysts with single MOx

(M =Y, Zr, Al, Ce and Mg). But the chemical and structural origin for such observed discrepancy of catalytic performance is discussed insufficiently. Authors claimed that the CeAlO_x/Ni/Ni-foam catalyst is characteristic of high O vacancy. We wonder whether this is mainly due to the dual-component CeO_x-AlO_x feature.

(3) Authors also claimed that CeO_x and AlO_x were existed in amorphous states separately. So, use of "CeAlO_x" to represent them seems not suitable.

(4) There are a few typos and grammar errors.

A point-by-point response

We have addressed all the comments point-by-point and revised the manuscript accordingly. In this response letter, comments from referees are in black typeface, and our responses are started with **Reply** typeface. All major changes have been highlighted in blue in the main text.

Reviewer #1 (Comments for the Author):

In this manuscript the authors report a nickel-based catalyst with remarkable catalytic activity, stability, and outstanding water resistance for CO₂ methanation, which also shows resistance to carbon deposition. The origin of ultrahigh thermal stability is mentioned, but the discussion on catalytic mechanisms is insufficient. Therefore, the work is still preliminary to publish as a qualified research. The authors need to improve their manuscript toward a high-level research.

Comment:

1. Some conclusions claimed by the authors are not supported by solid experimental data. This basically decreases the quality of this research. For example, the conditions which benefit the thermal stability, coke elimination and water resistance of the catalyst are explained by a model shown in Fig. 4c. This should be the critical novel point of this research, thus should be deeply discussed and revealed by probing the processes.

Reply: Thanks for your valuable comments. Deep discussion of the thermal stability, coke elimination and water resistance of the catalyst are shown in **Fig. R1**, and it also have been added in the revised manuscript (**Fig. 4e and Fig.5a-c**).

The excellent thermal stability of structural catalysts is tried to confirm by comparing the catalyst bed temperature rises under the same reaction condition (**Fig. R1a**), which reflects the degree of accumulation of hotspots on the catalysts. The temperature rise of the CeAlO_x/Ni/Ni-foam structure catalyst bed is limited 3 °C at a wide range of reaction temperature and CO₂ conversion, however, the CeAlO_x/Ni powder catalyst bed shows a obvious temperature rise of 20 °C, indicating that the temperature of hotspot on the structural catalyst bed is effectively diminished, thus preventing sintering of the catalytic active sites due to local temperature surge. Then, the bed pressure drop was performed to compare the mass transport efficiency of the catalysts under the same volume space velocity. The pressure drop of the CeAlO_x/Ni /Ni-foam structure catalyst

only increases of 0.2×10^5 Pa when the superficial velocity is as high as 300 mL/min, while the back pressure of powder catalyst increases to as high as 1.8×10^5 Pa at 300 ml/min (**Fig. R1b**). This enhanced mass transport efficiency prevents the accumulation of water in the catalyst bed, which probably contributes to the remarkable water resistance of the nickel foam-based catalyst (**Fig. R1c**).

Temperature programmed surface reaction (TPSR) for CH_4 decomposition was conducted on Ni/CeAlO_x and $\text{CeAlO}_x/\text{Ni}/\text{Ni-foam}$ catalysts, and $m/z=2$ (H_2) was monitored to indicate the methane cracking ($\text{CH}_4 = \text{C} + 2\text{H}_2$). In the CH_4 -TPSR experiment, the H_2 signal of Ni/CeAlO_x appears at 602 °C, when the nano-oxide was introduced on $\text{Ni}/\text{Ni-foam}$, the temperature of H_2 increases to 656 °C (**Fig. R1d**), suggesting that the nano-oxides tend to mask the highly uncoordinated sites of Ni which are the main active center for methane decomposition. This advantage accompanied with the much lower density of local hotspots in the Ni foam endows our catalyst excellent coke resistance.

Fig. R1 a. The comparison of temperature-rising for the Ni-foam-structured CeAlO_x/Ni catalyst and CeAlO_x/Ni catalyst; b. Pressure drop against N_2 gas superficial velocity. $\text{CeAlO}_x/\text{Ni}/\text{Ni-foam}$ (100 PPI), CeAlO_x/Ni (60-80 meshes); c. water resistance test of $\text{CeAlO}_x/\text{Ni}/\text{Ni-foam}$ catalyst (Reaction conditions: 240 °C, GHSV=10000 h⁻¹, $\text{CO}_2:\text{H}_2:\text{N}_2=18:72:10$, $P=0.1$ MPa); d.

temperature programmed surface reaction (TPSR) of methane on Ni/CeAlO_x and CeAlO_x/Ni/Ni-foam. Reaction conditions :10 vol% CH₄/Ar, GHSV=15000 h⁻¹.

2. In the conclusion part of this manuscript, the authors mentioned the adsorption/desorption behavior of CO₂ and hydrogen. However, the data about it is just a supplementary figure (Figure S7), which is not discussed in the main text.

Reply: Thanks for your comment. The discussion of the adsorption/desorption behavior of CO₂ and hydrogen in the conclusion part refers to the relative surface coverage of H₂ and CO₂. Compared to Ni/(Ce)AlO_x conventional catalysts, the reaction order of CO₂ increases from 0.04 (0.02) to 0.24 (0.21), and the reaction order of H₂ decreases from 0.82 (0.81) to 0.36 (0.34) (**Fig. R2**). The change of the apparent kinetic orders of H₂ and CO₂ suggests that the CO₂ coverage decreases while H₂ coverage is increases over the (Ce)AlO_x/Ni/Ni-foam inverse catalysts according to the Langmuir-Hinshelwood mechanism. Thus, the coverage of CO₂ and H₂ on the inverse catalysts get closer which is beneficial for the surface reaction. To avoid misunderstandings, we replace the adsorption/desorption behavior with the discussion of coverage in the revised manuscript.

Fig. R2 Reaction orders with respect to H₂ and CO₂ for methane formation.

3. It is said in the manuscript that oxygen vacancy is the active site, but the working mechanism is not well explained.

Reply: Thanks for your comment. The oxygen pulse experiments were conducted to determine the density of oxygen vacancies of CeAlO_x/Ni/Ni-foam and Al₂O₃/Ni/Ni-foam (as seen in **Method** in manuscript). The density of oxygen vacancies on CeAlO_x/Ni/Ni-foam and Al₂O₃/Ni/Ni-foam is 228 and 140 μmol/g_{cat}, respectively,

which is in good agreement with the deconvolution results of O 1s XPS spectra (Fig.R3a, also seen in Fig.S7). The analysis of CO₂ TPD profiles of CeAlO_x/Ni/Ni-foam and Al₂O₃/Ni/Ni-foam suggests the capacity of weak- and medium-adsorbed CO₂ are 150 and 102 μmol/g_{cat} (Fig. R3b). There is a good linear correlation between the capacity of weak and medium-adsorbed CO₂ and the density of oxygen vacancies (R²=0.98) (Fig. R3c). In addition, it is found that the intrinsic productivity of CH₄ at 160, 180, 200 and 220 °C presents linear dependence to the amount of weak and medium-adsorbed CO₂ (Fig. R3d). Therefore, the activity of CO₂ methanation is highly related with oxygen vacancies, which serve as the adsorption and activation sites of the reactive CO₂* species.

Fig. R3 a. In situ XPS spectra of O 1s of CeAlO_x/Ni/Ni-foam and Al₂O₃/Ni/Ni-foam catalysts; b. The CO₂-TPD profiles in varied Ni-base catalysts; c. Relationship between CO₂ adsorption capacity and amount of oxygen vacancies on CeAlO_x/Ni/Ni-foam and Al₂O₃/Ni/Ni-foam catalysts; d. The correlation of the STY methane (at 160 and 220 °C, respectively) and the amount of adsorbed CO₂ at 50-450 °C.

Reviewer #2 (Comments for the Author):

This manuscript presents a CeAlO_x/Ni/Ni-foam catalyst active for CO₂ methanation with exceptional stability against sintering and coking. To synthesize that material, the authors created a Ni(OH)₂ overlayer on Ni foam by urea-etching and deposited different metal oxides to create inverse structures. The conclusions are well supported by extensive characterization and thorough study of the stability of the materials. I very much enjoyed reading this elegant study, and I am confident it will be of interest to the readers of Nature Communications.

Below are some minor points that need attention:

Comment:

1. The font in most figures is too small. Difficult to read.

Reply: Thanks for your suggestion, the font size has been adjusted in the revised manuscript.

2. Page 4: “Ni supported on the Al₂O₃ and CeAlO_x oxide supports are prepared by the precipitation method (Ni loading is controlled at 13 wt%).” There is no mention to those catalysts before, so it might be confusing. I would recommend including a sentence explaining the motivation behind preparing those catalysts and further details about the synthesis.

Reply: Thanks for your suggestion. The sentence is modified as “Ni supported on the Al₂O₃ and CeAlO_x oxide supports are prepared by the precipitation method (Ni loading is controlled at 13 wt%) to compared with the inverse oxide/Ni composite catalysts and understand the importance of Ce doping” . The detailed synthesis method of control catalysts has also been added in **Supporting Information**.

3. Page 4: “···Ni-foam substrate (marked with cubs).” There are more symbols in that figure, so “marked with cubs” could be omitted.

Reply: Thanks for your valuable suggestion. We have revised it in the manuscript and highlighted in blue.

4. Page 4: The term “hysterescence loops” is not correct. It should be “hysteresis loops”.

Reply: Thanks for your valuable suggestion. We have revised it in the manuscript

and highlighted in blue.

5. Page 4: Instead of “...suggest that NiO nanoparticles (~5.4 nm) are deposited...,” it is recommended to use “...suggest that NiO nanoparticles (distribution centered at ~5.4 nm) are deposited...”

Reply: Thanks for your valuable suggestion. We have revised it and highlighted in blue.

6. Figure 2a: “Conversion of thermodynamic equilibrium” is too far from the dotted line, so the reader might get confused.

Reply: Thanks for your valuable advice. The thermodynamic equilibrium conversion curve has been removed from Fig. 2a.

7. Figure 2: It is recommended to use different symbols as well, not only rely on different colors.

Reply: Thanks for your valuable suggestion. The symbols in Fig. 2 have been modified in our revised manuscript, as shown in Fig. R4.

Fig. R4 Temperature-dependent a. CO₂ conversion and b. CH₄ selectivity of the CeAlO_x/Ni/Ni-foam, Al₂O₃/Ni/Ni-foam, CeO₂/Ni/Ni-foam, Ni/Al₂O₃, and Ni/Ni-foam catalysts (Reaction conditions: GHSV=10000 h⁻¹, 160-300 °C CO₂:H₂:N₂=18:72:10, P=0.1 MPa)

8. Figure 2d: CeO_x-Al₂O₃/Ni/Ni-foam shows a different nomenclature, better be consistent to avoid confusion.

Reply: Thanks for your suggestion. We have unified the catalyst as “CeAlO_x/Ni/Ni-foam” in the revised manuscript and highlighted in blue.

9. Page 5: “CeAlO_x hetero-structure in the catalyst” is unclear. Is that a mixed metal oxide?

Reply: Thanks for your suggestion. It is a mixed metal oxide based on characterizations. We have revised them to “mixed metal oxides” for a better understanding and highlighted in blue.

10. Page 5: “CO₂ adsorption” would be more accurate than “CO₂ capture.”

Reply: Thanks for your valuable suggestion. We have revised it as “CO₂ capture” in the revised manuscript and highlighted in blue.

11. Figure 3 legend: “after heating-cooling cycle tests” as there were more than one.

Reply: Thanks for your comment. We have revised it to “after heating-cooling cycle tests” in the revised manuscript and highlighted in blue. (the **Fig.3** is moved to **Fig.4**)

12. Page 7: Missing information about how the authors ensured they worked in the kinetic region.

Reply: Thanks for your comment. In our experiments, by varying the GHSV for different catalysts, we ensured that all the CO₂ conversion used to calculate of E_a were below 6% (**Fig. R5** and also seen in **Fig. S12**). A more detailed description has been added to **method** part.

Fig. R5 The profiles of CO₂ conversion as a function of reaction temperature.

13. Page 7: “inverse ensembles” might be difficult to understand. Suggestion: MO_x ensembles.

Reply: Thanks for your suggestion. We have revised it as “inverse MO_x/Ni composites” in the manuscript and highlighted in blue.

14. Page 7: “the coverage of CO₂ is reduced and H₂ surface coverage is intensified on the inverse ensembles” is difficult to read. Suggestion: “that the CO₂ coverage decreases and H₂ surface coverage is intensified over the MO_x ensembles.”

Reply: Thanks for your comment. We have revised it as “that the CO₂ coverage decreases and H₂ surface coverage is intensified over the MO_x ensembles” in the revised manuscript and highlighted in blue.

15. References: Authors could include some references about the benefits of using inverse catalysts.

Reply: Thanks for your valuable suggestion. The references regarding the benefits of using inverse catalysts have been added into the revised manuscript. (Ref 29: *Chem. Eng. J.* **2023**, 470, 144006; Ref. 30: *Nat Commun.* **2019**, 10, 3470; Ref. 31: *Appl. Catal. B* **2023**, 334, 122839; Ref. 32: *Appl. Catal., B* **2024**, 344, 123656)

16. SI, catalyst evaluation: Unclear what the authors mean with “a non-quartz tube.” Is that a stainless-steel tube?

Reply: Thanks for your valuable comment. “a non-quartz tube” is a typo, it is a quartz tube used for measurement. We have revised it in the Supporting Information and highlighted in blue.

17. SI, catalyst evaluation, equations: please define F, C, A, n, and V_{cat}.

Reply: Thanks for your kind comment. The definitions of F, C, A, n and V_{cat} have been added to the Supporting Information and highlighted in blue.

18. SI, catalyst characterization: missing details about ICP-AES.

Reply: Thanks for your kind comment. Detailed preparation methods of ICP-AES samples have been added in the **Method** part and highlighted in blue.

19. SI, STEM and EDX imaging: Using EDX and EDS in the text. Although it is the

same, better use one to be consistent.

Reply: Thanks for your kind comment. We have revised all to “EDS” in both manuscript and supporting information.

20. Figure S1, legend: Also NiO/Ni-foam and Ni/Ni-foam.

Reply: Thanks for your kind comment. We have corrected the legend and highlighted in blue.

21. Figure S2: Used NF instead of “Ni-foam”. Better be consistent to avoid confusion.

Reply: Thanks for your kind comment. We have revised it as “Ni-foam” in the revised manuscript.

22. Table S1: There is a typo. It should read as “CeAlO_x/Ni/Ni-foam-used.”

Reply: Thanks for your kind comment. We have revised it in the revised Table S1.

23. Figure S9: It is recommended to use different symbols as well, not only rely on different colors.

Reply: Thanks for your kind comment. The figure has been changed as shown in **Fig. R6**. (the **Fig.S9** is moved to **Fig.S10**)

Fig. R6 Temperature-dependent activities of Al₂O₃/Ni/Ni-foam catalysts with different promoters. Reaction conditions for the catalytic test: GHSV=10,000 h⁻¹, 160-320 °C, CO₂:H₂:N₂=18:72:10, P = 0.1 MPa.

24. Figure S10: Lack of details about working in the kinetic region. Different GHSVs and temperatures.

Reply: Thanks for your kind comment. Details of temperature and GHSV have been added into supporting information (**Fig. R7** and **Tab. R3**), which used to ensure the CO₂ conversion was less than 15% when the catalyst is working in kinetic region (Ref. 44: ACS Catal. **2022**, 12, 17, 10587–10602). **Fig R7** and **Tab. R3** are added in to supporting information as **Fig. S11** and **Tab. S4**.

Fig. R7 Temperature-dependent activities of CeAlO_x/Ni/Ni-foam and Ni/CeAlO_x catalysts with different promoters. Reaction conditions for the catalytic test: CeAlO_x/Ni/Ni-foam: GHSV=60,000 h⁻¹, 160-200 °C, Ni/CeAlO_x: GHSV=10,000 h⁻¹, 160-200 °C

Table R3 Comparison of activity of CeAlO_x/Ni/ Ni-foam and Ni/CeAlO_x catalysts in kinetic region.

Catalyst	Temperature (°C)	GHSV (h ⁻¹)	X _{CO2} (%)	S _{CH4} (%)	CH ₄ .STY (mmol/mL _{cat} /h)
CeAlO _x /Ni/Ni-foam	200	60000	10.6	100	65.4
Ni/CeAlO _x	200	10000	4.2	100	4.3

25. Table S4: Very diluted streams. It is assumed that is to get low CO₂ conversion (kinetic regime), but that's not mentioned in the main text.

Reply: Thanks for your comment. Detailed description of the application of diluted streams has been added to the revised manuscript and highlighted in blue. The purpose of diluting CO₂ reaction gas is to convert CO₂ in the kinetic region while also eliminating the effect of hotspot.

26. Table S5: R_{CH₄} seems to indicate productivity. The use of “R” might be confusing.

Reply: Thanks for your kind comment. “R_{CH₄}” in supporting information has been changed to “CH₄-STY” to better represent space time yield of CH₄.

Reviewer #3 (Comments for the Author):

This work deals with development of a new-type Ni-foam-structured inverse CeAlO_x@Ni nanocomposite catalyst for application in the CO₂ methanation reaction. The results indicated that this approach worked effectively and efficiently to couple high activity/selectivity and promising stability with enhanced heat/mass transfer and high permeability. I recognized the relevance of the subject and the approach used by the authors is appropriate. In principle, I suggest accepting this draft for publication after MAJOR revision.

Comment:

1. Inverse structure of CeAlO_x@Ni nanocomposite is not convincing, at least the evidences in the current manuscript is insufficient. High-quality images with clear inverse structure at interfaces between Ni particles and CeAlO_x ensembles.

Reply: Thanks for your suggestion. The high resolution HAADF-STEM images of CeAlO_x@Ni nanocomposite have been represented **Fig. R8**. It can be seen that two kinds of lattice fringes of 0.265 nm and 0.209 nm coexist on the catalyst, which corresponds to CeAlO₃ (PDF#81-1186;110) and NiO (PDF#89-7131;111). The CeAlO_x ensembles supported on Ni support can be observed. **Fig. R8a** has been added in to the manuscript as **Fig. 1f**, and **Fig. R8b** has been added as **Fig. S5**.

Fig. R8 High-resolution aberration-corrected HAADF-STEM images of CeAlO_x/Ni/Ni-foam catalyst. a. HR-STEM images of the CeAlO_x/Ni/Ni-foam catalyst; b. The analysis of lattice fringes of the HR-STEM images.

2. The CeAlO_x/Ni/Ni-foam catalyst is superior over the other MO_x/Ni/Ni-foam catalysts with single MO_x (M = Y, Zr, Al, Ce and Mg). But the chemical and structural origin for such observed discrepancy of catalytic performance is discussed insufficiently. Authors claimed that the CeAlO_x/Ni/Ni-foam catalyst is characteristic of high O vacancy. We wonder whether this is mainly due to the dual-component CeO_x-AlO_x feature.

Reply: Thanks for your comments. The corresponding adsorption capacity of weak- and medium-adsorbed CO₂ (<450 °C) of CeAlO_x/Ni/Ni-foam, Al₂O₃/Ni/Ni-foam, Ni/Al₂O₃ and Ni/CeAlO_x are determined to 150, 102, 15 and 70 μmol/g_{cat} (**Fig. R9a**), which show linear correlation with the intrinsic productivity of CH₄ at 160, 180 and 200 °C on each catalyst (Ref. 44: ACS Catal. **2022**, 12, 17, 10587–10602) (**Fig. R9b**). The density of oxygen vacancies on CeAlO_x/Ni/Ni-foam and Al₂O₃/Ni/Ni-foam were additionally measured by oxygen pulse chemisorption experiments (**Fig. R9c**), which are 228 and 140 μmol/g_{cat}, respectively, which suggests that the introduction of Ce increases the density of O_v in the mixed oxides. According to the reference, the surface oxygen vacancies are the similar sites for CO₂ adsorption at weak and medium strength. Then, the relationship between the weak- and medium-adsorbed CO₂ and oxygen vacancies is also established for our work, which appears with excellent linear relationship (R²=0.98) like previous reports (**Fig. R9c**). Therefore, the activity of CO₂ methanation at low temperature is highly related with oxygen vacancies. The introduction of Ce forms CeAlO_x mixed oxide, which enhances the generation of oxygen vacancies (based on

Raman, XPS and HR-HAADF-STEM characterization. See detailed explanation seen in the following Reply) and further promotes the activity of methane.

To further clarify the enhanced performance effect of mixed-oxide/Ni inverse configuration, in situ DRIFTS is carried out under typical CO₂ hydrogenation conditions (Fig. R10a-b). Under the reaction atmosphere (CO₂ + 4H₂), bridged CO* (1833 and 1930 cm⁻¹), formate (2970, 1563-1580, 1380 cm⁻¹) and methoxy (2845, 2926 cm⁻¹) species are observed on CeAlO_x/Ni/Ni-foam catalyst. In contrast, only formate and methoxyl species are observed on Al₂O₃/Ni/Ni-foam catalyst. In addition, it is observed that when the CO₂ in the feed is cut off, CO* and formate species on the CeAlO_x/Ni/Ni-foam catalyst are rapidly consumed together with the formation of methane (Fig. R10c-f). Similarly, the consumption of formate species and formation of methane are observed on the Al₂O₃/Ni/Ni-foam catalyst when the CO₂ is cut off. These results indicate that CO₂ methanation follows the formate path and CO* path on the CeAlO_x/Ni/Ni-foam catalyst, while only CH₄ formate path could occur on the Al₂O₃/Ni catalyst. Therefore, these two possible reaction pathways synergistically promote the lower temperature methanation on the CeAlO_x/Ni/Ni-foam catalyst.

The relevant content has been added to the revised manuscript.

Fig. R9 a. The CO₂-TPD profiles in varied Ni-base catalysts; b. The correlation of the STY methane (at 160 and 220 °C, respectively) and the amount of adsorbed CO₂ at 50-450 °C; c. Relationship between CO₂ adsorption capacity and amount of oxygen vacancies on CeAlO_x/Ni/Ni-foam and Al₂O₃/Ni/Ni-foam catalyst

Fig. R10 Surface intermediates investigation by in situ DRIFTS characterization. a-c. In-situ DRIFTS spectra of the CO₂/H₂ reaction on CeAlO_x/Ni/Ni-foam and Al₂O₃/Ni/Ni-foam catalyst. The catalysts are exposed to 80% H₂/20% CO₂ (10 mL min⁻¹) atmosphere at 180 °C for 90 min; b-d. In-situ DRIFTS spectra of the H₂ atmosphere on CeAlO_x/Ni/Ni-foam and Al₂O₃/Ni/Ni-foam catalyst (pretreat 90 mins in 80% H₂/20% CO₂ atmosphere at 180 °C and the inlet is switched to 80% H₂/20% Ar and maintained at the same temperature for 90 min); e. normalized intensities of the typical surface species, formate (~1565 and 1380 cm⁻¹ for CeAlO_x/Ni/Ni-foam; ~1572 cm⁻¹ for Al₂O₃/Ni/Ni-foam) species versus reaction time. f. DRIFTS results of the CeAlO_x/Ni/Ni-foam catalyst and Al₂O₃/Ni/Ni-foam catalyst in the stream of CO₂/H₂ mixture under 0.1 MPa respectively at 180 °C.

3. Authors also claimed that CeO_x and AlO_x were existed in amorphous states separately. So, use of “CeAlO_x” to represent them seems not suitable.

Reply: Thanks for your comment. In the XRD characterization, the loaded oxides on the NiO are invisible, it is hard to determine whether it is mixed oxide or not. However, the formation of CeAlO_x mixed oxides is confirmed by HR-HAADF-STEM, Raman spectroscopy, and quasi in-situ XPS characterization, as shown in **Fig. R11** (also seen in **Fig. 1f, 1g, and 1i**).

As shown in **Fig. R11a-b**, the lattice fringe of 0.265 nm is present on the CeAlO_x/NiO catalyst, which is according to the (110) plane of CeAlO₃. Then, Raman spectroscopy is applied to investigate the structure (**Fig. R11c**). No typical peak of CeO₂ at 464 cm⁻¹ was resolved on CeAlO_x/NiO, indicating that separate CeO₂ phase does not exist in the catalyst. Meanwhile, it can be seen that the peak at 574 cm⁻¹ of Ni-O-Al on AlO_x/NiO catalyst shifted toward the higher frequency of 605 cm⁻¹ of AlO_x/NiO, which is due to the formation of Al-O-Ce³⁺ bonds. Also, in the in-situ XPS profiles, the binding energy of Al 2p in the AlO_x/Ni is 74.0 eV, while the Al 2p in CeAlO_x/Ni shows lower binding energy at 73.7 eV, further confirming the formation of CeAlO_x mixed oxide and Ce donates electron to Al (**Fig. R11d**). Therefore, the formation of CeAlO_x structure can be reasonably confirmed, and thus “CeAlO_x” was used to represent the mixed oxides. The “corresponding to the presence of an amorphous phase” in the manuscript was deleted to avoid misunderstanding.

Fig. R11 a. HR-STEM images of the $\text{CeAlO}_x/\text{NiO}/\text{Ni-foam}$ catalyst; b. The analysis of lattice fringes of the HR-STEM images; c. Raman spectra of the $\text{NiO}/\text{Ni-foam}$, $\text{Al}_2\text{O}_3/\text{NiO}/\text{Ni-foam}$, $\text{CeAlO}_x/\text{NiO}/\text{Ni-foam}$, CeO_2 and Al_2O_3 catalysts; d. In-situ XPS of Al 2p of $\text{CeAlO}_x/\text{NiO}/\text{Ni-foam}$, $\text{CeAlO}_x/\text{Ni}/\text{Ni-foam}$, $\text{Al}_2\text{O}_3/\text{NiO}/\text{Ni-foam}$ and $\text{Al}_2\text{O}_3/\text{Ni}/\text{Ni-foam}$ catalysts.

4. There are a few typos and grammar errors.

Reply: Thanks for these valuable comments, the English language has been revised in the manuscript.

REVIEWER COMMENTS

Reviewer #1 (Remarks to the Author):

The authors have conducted more experiments and characterizations to support the enhanced activity of CeAlO_x/Ni catalyst for CO₂ hydrogenation. The paper might be considered to be accepted for publication on Nature Communications after addressing some issues.

1) The authors can discuss the effects of metal-oxide interface in enhancing metal-support interaction and CO₂ methanation by referring to Nat. Commun. 2022, 327. The metal-support interface is critical for CO₂ hydrogenation.

2) The authors argued the appearance of CeAlO₃ (PDF#81-1186;110) by measuring lattice fringes. This is not a convincing support owing to the errors in measurement. And there is no CeAlO₃ peak in the XRD pattern.

3) Fig. 5d is confusing. Why are Ni and CeAlO_x shown in the same color? The structures are not clearly labeled.

4) The catalyst is mainly made of Ni foam, thus should be very magnetic. How did the authors perform the TEM characterization? Is it possible to show some TEM images of a lower scale?

Reviewer #2 (Remarks to the Author):

The authors properly addressed all my comments, so I recommend this manuscript for publication. I would like to note that eq. (5) in the SI is the same as eq. (4), so that should be deleted.

Reviewer #3 (Remarks to the Author):

The author gives better answers for most of the reviewer's questions. In the revised version, most problems in the comments before have been explained or answered, so in this case, this manuscript can be recommended for publication with minor modification. It is suggested to replace "ultra-high" in the title and main texts with word like "promising".

A point-by-point response

We have addressed all the comments point-by-point and revised the manuscript accordingly. In this response letter, comments from referees are in black typeface, and our responses are started with **Reply** typeface. All major changes have been highlighted in **blue** in the main text.

Reviewer #1 (Comments for the Author):

The Reviewer thought that “The authors have conducted more experiments and characterizations to support the enhanced activity of CeAlO_x/Ni catalyst for CO₂ hydrogenation. The paper might be considered to be accepted for publication on Nature Communications after addressing some issues.”

Comment:

1. The authors can discuss the effects of metal-oxide interface in enhancing metal-support interaction and CO₂ methanation by referring to Nat. Commun. 2022, 327. The metal-support interface is critical for CO₂ hydrogenation.

Reply: Thanks for your valuable comments. The discussion of the oxide-metal interaction in enhancing the stability of catalyst and prevent the sintering of Ni substrate has been added in the manuscript referring the reference recommended by the reviewer (Ref.48 Nat. Commun. 2022, 327) in the following sentences. “This phenomenon implies that the interaction between the oxide and Ni substrate effectively inhibit the migration of Ni species and thereby prevent the undesirable sintering.”

2. The authors argued the appearance of CeAlO₃ (PDF#81-1186;110) by measuring lattice fringes. This is not a convincing support owing to the errors in measurement. And there is no CeAlO₃ peak in the XRD pattern.

Reply: Thanks for your comment. As seen in STEM, the particle size of oxides are ~2nm, which is undetectable based on the XRD technique. The aberration-corrected high-resolution HAADF-STEM image displays the CeAlO_x lattice spacing of 0.265 nm, which falls between the lattice spacing of the (111) plane of CeO₂ (d=0.315 nm; PDF# 75-0076) and that of the (104) plane of Al₂O₃ (d=0.252 nm; PDF# 85-1337), suggesting the formation of CeAlO_x mixed oxides. The difference is sufficiently large so that the

measuring error can be excluded. Furthermore, the formation of CeAlO_x mixed oxides is confirmed by Raman spectroscopy and quasi in-situ XPS characterization, as shown in Fig. R1.

Fig. R1 a. Raman spectra of the $\text{NiO}/\text{Ni-foam}$, $\text{Al}_2\text{O}_3/\text{NiO}/\text{Ni-foam}$, $\text{CeAlO}_x/\text{NiO}/\text{Ni-foam}$, CeO_2 and Al_2O_3 catalysts; b. In-situ XPS of Al 2p of $\text{CeAlO}_x/\text{NiO}/\text{Ni-foam}$, $\text{CeAlO}_x/\text{Ni}/\text{Ni-foam}$, $\text{Al}_2\text{O}_3/\text{NiO}/\text{Ni-foam}$ and $\text{Al}_2\text{O}_3/\text{Ni}/\text{Ni-foam}$ catalysts.

3. Fig. 5d is confusing. Why are Ni and CeAlO_x shown in the same color? The structures are not clearly labeled

Reply: Thanks for your valuable comment. It has been revised in the Fig. 5d, as shown in Fig. R2.

Fig. R2 Schematic representation of a Ni-foam skeleton constrained stabilized inverse nickel catalyst and a reference sample

4. The catalyst is mainly made of Ni foam, thus should be very magnetic. How did the

authors perform the TEM characterization? Is it possible to show some TEM images of a lower scale?

Reply: Thanks for your suggestion. The low scale TEM images of the CeAlO_x/Ni/Ni-foam catalysts are shown in Figure R3 (added as Fig. S4 in the revised SI), in which the foam structure can be resolved clearly. In TEM characterization of inverse structure, the active phase CeAlO_x/Ni was scraped from the passivated nickel foam catalyst (after reduction) and used for sample preparation. This method is frequently used to avoiding the interference of strong magnetic properties of nickel foam substrate on high resolution TEM imaging (Ref.xx Adv. Mater. 2021, 33, 2005587). The detailed preparation method of TEM sample has also been added in Supporting Information.

Fig. R3 a Low-magnification TEM image for CeAlO_x/Ni/Ni-foam catalyst; **b** enlarged TEM image for CeAlO_x/Ni/Ni-foam catalyst.

Reviewer #2 (Comments for the Author):

The reviewer thought that “properly addressed all comments” and recommend this manuscript for publication

Comment:

1. I would like to note that eq. (5) in the SI is the same as eq. (4), so that should be deleted.

Reply: Thanks for your valuable advice. The eq. (5) has been removed from supporting information.

Reviewer #3 (Comments for the Author):

Review #3 thought that “The author gives better answers for most of the reviewer's questions. In the revised version, most problems in the comments before have been explained or answered” and suggested “this manuscript can be recommended for publication with minor modification”.

Comment:

1. It is suggested to replace "ultra-high" in the title and main texts with word like "promising".

Reply: Thanks for your valuable suggestion. We have revised it in the manuscript and highlighted in blue.

REVIEWERS' COMMENTS

Reviewer #1 (Remarks to the Author):

This manuscript is acceptable for publication in Nat Commun as it is.

A point-by-point response

We have addressed all the comments point-by-point and revised the manuscript accordingly. In this response letter, comments from referees are in black typeface, and our responses are started with **Reply** typeface. All major changes have been highlighted in **blue** in the main text.

Reviewer #1:

This manuscript is acceptable for publication in Nat Commun as it is.

Reply: We appreciate your support and affirmation.